# Effectiveness of Utilizing Remote Sensing and GIS Techniques to Estimate the Exposure to Organophosphate Pesticides Drift over Macon, Alabama

Gamal El Afandi *, Hossam Ismael , Souleymane Fall and Ramble Ankumah

College of Agriculture, Environment and Nutrition Sciences, Tuskegee University, Tuskegee, AL 36088, USA;
hismael@tuskegee.edu (H.I.); rankumah@tuskegee.edu (R.A.)
* Correspondence: gelafandi@tuskegee.edu; Tel.: +1-334-724-4790

**Abstract:** Farmers utilize pesticides extensively on their farms to control weeds and insects, as well as increase crop productivity. Despite these advantages, their excessive use poses a serious threat, particularly to the population living at the nexus of urban and rural areas. Exposure to pesticide drift can be investigated using geospatial tools. Remote sensing technology and Geographic Information Systems (GIS) techniques have been used intensively and constitute trusted tools in different sectors, especially in agriculture. Remote sensing depends on processing the electromagnetic radiation reflected and emitted from the ground target and can be used to identify the main units of Land Use Land Cover (LULC), in addition to measuring crop areas exposed to pesticides. GIS has powerful tools for building a spatial geo-database of pesticide exposure drift. Therefore, the major objective of the research was to explore the effectiveness of using remote sensing and GIS techniques to estimate the exposure organophosphate pesticides drift over Macon County, Alabama. To achieve this objective, the Cropland Data Layer (CDL) dataset, the available pesticide usage data, and gridded population data were used to estimate the potential pesticide drift on the Macon County level. In addition, the AgDRIFT model was used to estimate the potential drift of pesticides from their intended targets at the field level. The results indicated that 6.6% of Macon County's residents are considered potentially severely exposed, and the potentially affected population resides primarily in rural areas. In comparison, 23% of residents of the urban-rural interface are considered to have potentially medium to high exposure. In addition, 38% of residents living in suburban areas are considered to have potentially low-to-medium exposure. The results indicated that both GIS and remote sensing could play an effective role in estimating pesticide exposure drift at the State or County level. In addition, the AgDRIFT model was more appropriate for estimating pesticide drift at the field level.

**Keywords:** agricultural pesticides; spray drift; AgDRIFT

## 1. Introduction

Chemical treatments known as pesticides are frequently utilized in agriculture. These substances work to eliminate undesirable organisms, including fungi, weeds, and insects, that could otherwise ruin crops or limit their productivity. Other sectors also use pesticides to destroy exotic plants, remove weeds and shrubs from roads, and limit the formation of algae in water bodies. Pesticides have been heavily utilized in agriculture all over the world to guarantee higher crop yields [1–3]. According to the World Atlas 2022, the United States of America (USA) uses about 386.000 kg of pesticides annually, making it the second-largest consumer after China [4]. Herbicides account for 40% of pesticide use globally, followed by insecticides at 17% and fungicides at 10% [5]. They are naturally poisonous, making up one of the most dangerous classes of pollutants for flora, animals, and the environment [6].

It is confirmed that exposure to pesticide fumes in the atmosphere after the chemical has been applied by farmers is responsible for some of the health problems affecting

populations living in the urban-rural interface. However, lacking exposure data frequently imposed limitations on these examinations and experiments [6,7]. Short- and long-term exposure to pesticides can cause congestion of the heart, lungs, and kidneys, weight loss, and reproductive effects. Children are predominantly vulnerable to the impacts of pesticide drift due to their age, weight, and development [7]. Previous health studies on pesticide drift have established a relationship between exposure to pesticides in residential areas near agricultural fields and the high number of childhood cancers and brain tumors [8–10]. Moreover, exposure to pesticides is associated with adverse reproductive and neurologic/neurobehavioral effects and thyroid dysfunction [11–14].

Pesticide drift refers to the off-target movement of pesticide droplets during and after treatment. The coarse droplets move over short distances and fall near the launch point. Fine particles can remain suspended in air currents for prolonged periods and be carried away from the target areas to their adjacent ones. As a result, pesticide drift has become an increasingly debatable issue at the urban-rural margins and affects social justice, especially concerning quality of life [15]. It is widely confirmed that farmers and communities surrounding agricultural areas are the most at-risk groups for pesticide drift. However, applicators are always the most vulnerable to pesticide application due to insufficient protection, droplet drift, leaks, and other defects in spraying equipment [16]. Wind speed, nozzle type, spray pressure, and relative humidity are the key variables that affect spray drift. However, variables like wind and application methods may increase the possibility of pesticides drifting from their intended target. Pesticide drift exposure is usually classified into two categories based on the type of application: occupational and non-occupational exposure drift. Occupational exposure drift occurs during pesticide spraying in pesticide users and their immediate family members, farmers, and applicators [17].

While non-occupational pesticide exposure occurs in residential areas and around water bodies near pesticide application locations, where chemicals drift from 50 m to 2 km within residential areas and human activity centers, depending on the type of treatment, whether ground, orchards, or aerial application. Despite the significant impacts of pesticide exposure drift on population and health conditions, there have only been a few research studies that have measured non-occupational exposure drift; this is due to being constrained by detailed pesticide source data and measuring drift exposure applications on a local scale [18,19].

To estimate organophosphate pesticides drift, it is vital to acquire detailed data concerning the pesticide labels and application type and to determine the buffer zone between the cropland and the nearest residential area. This study is primarily concerned with measuring non-occupational exposure drift because its negative effects on residential areas, sensitive crops, and water bodies are greater than those from occupational exposure. Therefore, estimating organophosphate pesticide exposure drift at the county level is of utmost importance from meteorological, health, and environmental perspectives.

Monitoring the concentrations of pesticides in the atmosphere is one of the research issues worth studying due to the significant impact on the drift of pesticides from their target, and modern techniques face many difficulties in this type of complex study. It is recognized that satellite images are used to determine land use and land cover due to the dependence on those satellites for monitoring and measurement at different radiometric, spectral, spatial, and temporal resolutions. These satellite images can be obtained free of charge from the United States Geological Survey website (USGS) (https://earthexplorer.usgs.gov/, accessed on 20 March 2023). Landsat satellite images allow researchers to create an innovative opportunity to study the impact of exposure to pesticide drift from the target, especially in agricultural areas where ground measurements are not available [20,21].

Controlling studies concerning pesticides and human beings is far more complex due to the myriad factors that influence human disease and health. There are countless classes and types of pesticides on the market, all of which interact with organisms differently, making studies on pesticides all the more daunting due to the massive number of pesticides on the market and the lack of a database of organophosphate pesticide uses [5,6].

It is worth mentioning that we found some studies around the USA that described ways to analyze exposure to environmental pesticides using modern techniques. They mainly monitored the possibility of using modern techniques to calculate quantities of agricultural pesticides applied in the vicinity of a residential site as a proxy for non-food exposures. Marusek [22] and Ritz [23] focused on the surrounding agricultural land using GIS. Nuckols [24], Ward [25], and Teysseire [26] used the dwelling distance from the fields, but they considered the possibility of obstacles such as forests and trees.

Previous studies have utilized remote sensing and GIS techniques to study the impact of pesticide exposure drift on the community and environment, particularly in counties without a record of pesticide use. Hence, their methods and findings will be more beneficial at the county level, such as in Macon County. Dappen [27] used the supervised classification technique using GIS to create a land use land cover map. It was created on a large scale for the 2005 growing season so that it could be developed to include the entire state of Nebraska. Moreover, Wang [28] created a model for assessing the potential spray drift in off-croplands. This study concluded that spray drift could be simulated and estimated according to the application type and the buffer zone range from the unintended targets.

Notably, previous research supported the idea that pesticide exposure drift might be estimated by combining population and land use land cover data with standard pesticide usage data. Wan [29] provided an overview of potential pesticide exposure without detailed pesticide usage data in Nebraska. His methodology examines the acreage of corn and soybeans within a 450-m buffer zone from a population centroid to determine the possible population in Madison County exposed to pesticide drift. Wan utilized county-wide USGS pesticide usage estimates, which are calculated by a survey. He used census data, available pesticide data, and land use land cover data. He overlaid them all as layers in GIS. Wan conducted a weighted pesticide exposure study and found that 12% of Nebraska residents were at risk due to exposure to pesticide drift. Brouwer [30] created a spatiotemporal model to estimate potential environmental exposure to individual pesticides in the Netherlands as part of an ongoing case-control study on Parkinson's disease. Anna Myers [31] assessed the potential population in Madison County exposed to pesticide drift, which is accomplished by examining the amount of cotton, corn, and soybean within a 450-m buffer zone from a population centroid. Despite previous studies having disclosed an association between pesticide exposure drift on health issues and the environment, however, they are limited by pesticide data source and exposure technique. These limitations are exacerbated in states that do not have a pesticide reporting database, such as Alabama. A quick inspection of previous studies reveals that there are no attempts to conduct studies related to an estimate of the agricultural pesticides' exposure drift over Macon, Alabama, particularly in the sense of the concurrent rural and urban population exposure to pesticide drift.

Overall, based on available studies, it can be noted that most of them were not spatially evenly distributed in the United States, and many US states did not receive adequate study, despite the significance of understanding how pesticide use affects residential areas, applicators, and the environment. This is a result of the scarcity of pertinent data, particularly for the state of Alabama, which is needed for further study. Recalling all these challenges, the utilization of agricultural pesticides in Alabama has raised the risk of pesticide drift, particularly when more people live on agricultural land and are exposed to more chemicals. According to estimates, Macon County used 3.89 kg/ha of pesticides in 2021, which is more than the global average of 2.24 kg/ha [32,33].

Therefore, the major purpose of this study is to assess the effectiveness of utilizing remote sensing and GIS techniques to estimate the potential exposure to organophosphate pesticides drifting over Macon, Alabama. The current study focuses on answering the following questions: (1) how could pesticide exposure drift be estimated at the county and field level using data on land use, land cover, population grids, and available pesticide usage? And (2) how to upscale such results to serve decision-makers in the state of Alabama.

## 2. Materials and Methods

### 2.1. Study Area

Figure 1 depicts the location of Macon County, which is in east-central Alabama. Geographically, Macon County is situated between latitudes 32°7′49″ and 32°35′7″ N and longitudes 85°30′48″ and 86°1′19″ W. The county has a total area of 392,830 acres or roughly 650 Square miles when water bodies are included. Macon County shares borders with Montgomery County to the west, Elmore County to the northwest, Tallapoosa County to the north, Lee County to the northeast, Russell County to the east, and Bullock County to the south. Eighteen thousand four hundred thirty-nine people were living in Macon County in 2020 [34]. The largest city, Tuskegee, has 9395 residents [35].

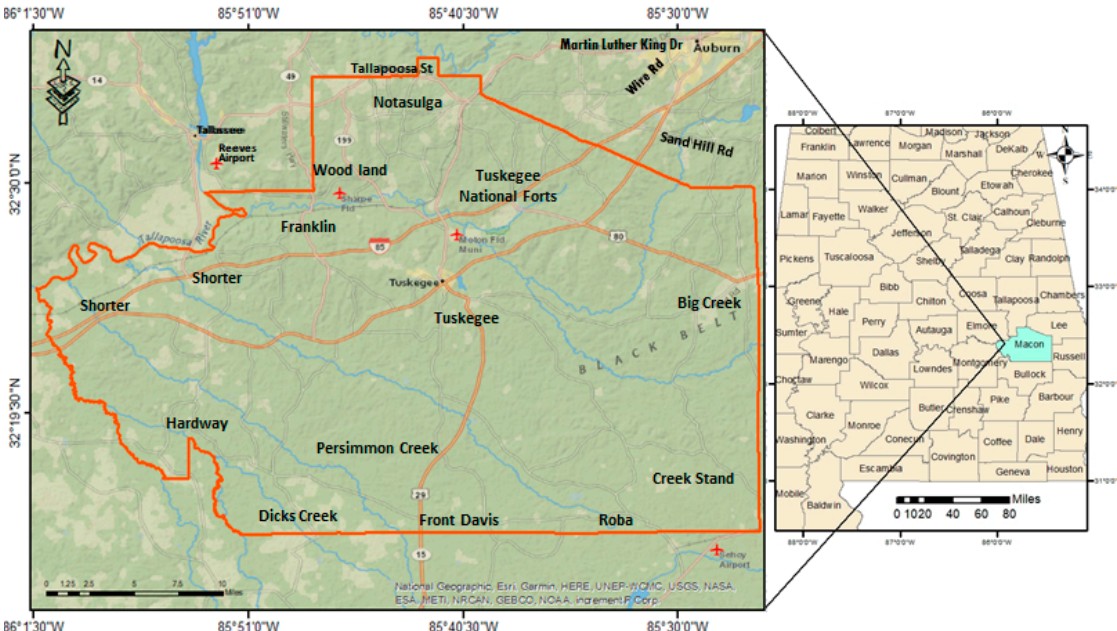

**Figure 1.** The location of Macon County, Alabama.

There are other minor communities in the County, including the incorporated cities of Notasulga, Franklin, and Shorter. Due to the continual presence of humid tropical air from the Gulf of Mexico, Macon County experiences lengthy, hot summers. The winters are cold and rather short. An unusual cold wave lasts for one or two days. The entire year sees a considerable amount of precipitation, and extended droughts are uncommon. For the growth of all crops, summer precipitation, primarily in the form of thunderstorms, is typically sufficient [36]. Twisters and other severe local storms occasionally occur in or close to the county. Every few years, a tropical depression or hurricane remnant that has gone inland causes unusually heavy rains for one to three days in the summer or fall [35,36].

Cropland and pastureland cover around 109,800 acres, or 23%, of the total land area in Macon County. In Macon County, where cotton is regarded as the primary crop grown, small grains and legumes are being sown as a winter cover crop by certain cotton farmers. Additionally, grown on smaller plots of land are soybeans, vegetables, and corn, which are largely used for silage [37]. Consequently, this study focuses on the most used agricultural pesticides in Alabama. Moreover, this study focuses on cotton and corn because they account for the majority of crop acreage, covering 26,305 acres. The pesticide data has been obtained from the USGS website under the Pesticide National Synthesis Project [38]. This study intended to determine what areas of Macon County might be at higher risk of exposure using remote sensing and GIS techniques.

The current study relied on evaluating pesticide exposure risk by overlaying LULC data with the pesticide usage data and gridded population data. Accordingly, this study was based on three prime inputs: land use, land cover, the available amount of pesticide,

and gridded population data for Macon County. Figure 2 provides a comprehensive picture of the study dataset and methodology for estimating the agricultural pesticide exposure drift over Macon County.

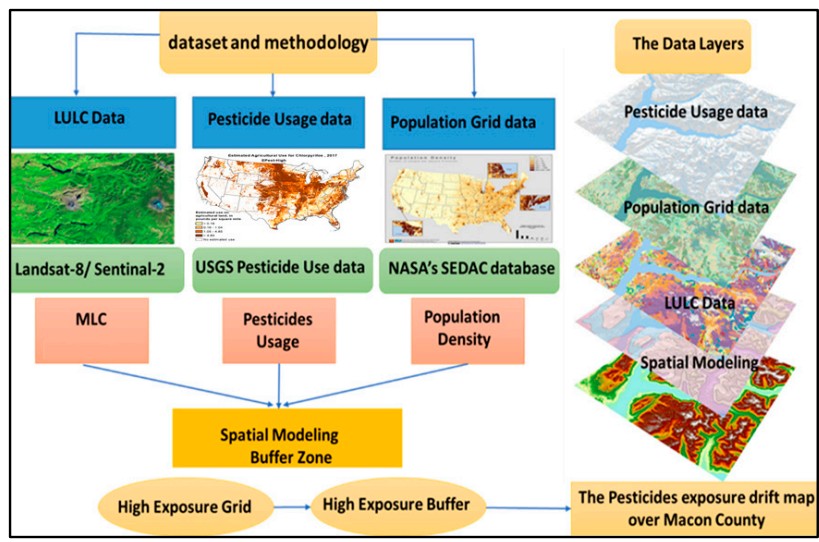

**Figure 2.** The study dataset and methodology.

## 2.2. The Data

### 2.2.1. Cropland and LULC Data

The cropland data for Macon County has been downloaded from the CropScape-Cropland Data Layer (CDL) dataset, which was used to identify specific crop areas. It is generated by the National Agricultural Statistical Service of the USDA (https://nassgeodata.gmu.edu/CropScape/, accessed on 2 April 2023). The CDL is processed using the Albers Equal-Area Conic Projection with a spheroid of GRS 1980 and a datum of NAD83. It is a high-resolution (30 m) satellite imagery with temporal coverage spanning more than 24 years (Figure 3) [39]. The main LULC units (residential areas, water bodies, cropland, and frosts) were derived from Landsat-8 OLI Table 1 and Sentinal-2 images using Maximum Likelihood Classification (MLC) method.

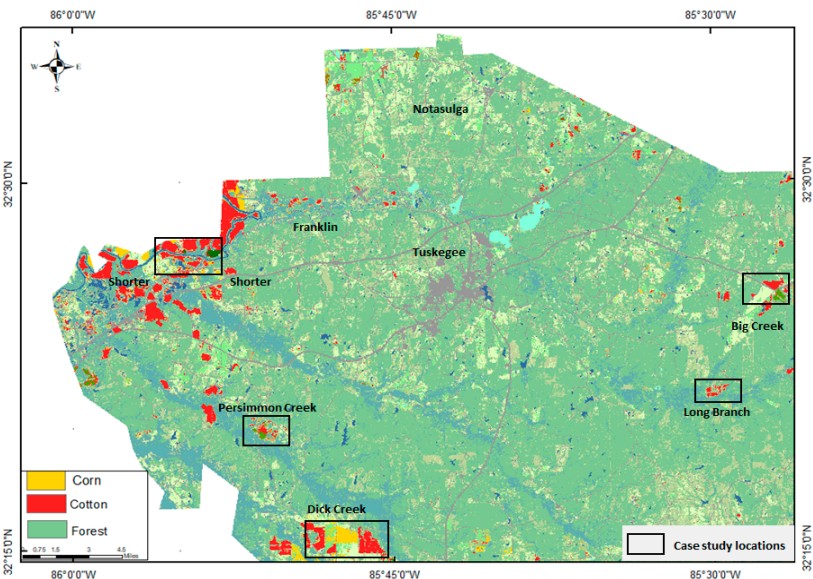

**Figure 3.** The cropland layer data over Macon County, 2017. Source: CropScape-Cropland Data Layer project.

**Table 1.** Specifications of Landsat-8 images.

| Image Number | | Date of Acquisition | Time of Acquisition (AM) | Cloudiness % |
|---|---|---|---|---|
| Row | Path | | | |
| 038 | 019 | 6 August 2017 | 10:23 | 6.04 |
| 037 | 020 | 6 August 2017 | 10:24 | 5.74 |

Source: https://www.usgs.gov/landsat-missions/landsat-8, accessed on 21 April 2023.

The optoelectronic multispectral sensor on Sentinal-2 allows for surveying with a resolution of 10 to 30 m in the visible, near-infrared (VNIR), and short-wave infrared (SWIR) spectral zones, with 13 spectral channels. Sentinal-2 images were retrieved via the Copernicus Open Access Hub https://scihub.copernicus.eu/, accessed on 31 March 2023.

2.2.2. Pesticide Usage Data

The annual agricultural pesticide database at the (USGS) is displayed in Tables 2 and 3 to specify the amount of data on pesticide consumption. Pesticide use reporting data from both the U.S. Environmental Protection Agency (USEPA) and the Department of Agriculture (USDA) National Agricultural Statistics Service (NASS) collects annual data on the sale and use of pesticides at the state level [39,40].

**Table 2.** The pesticide usage in Macon County compared with Alabama, 2017.

| Items | Definition | Alabama | Macon | % |
|---|---|---|---|---|
| ATLAS_ACRE | Area of the administrative region in acres | 3,241,319 | 389,636 | 1.20 |
| ACRES_HC | Acres of harvested cropland in acres | 2,205,766 | 26,305 | 1.19 |
| SQ_MI_HC | Square miles of harvested cropland | 50,647 | 41.00 | 0.1 |
| AGG_HI_KG | The sum of the EPest-high estimates for all seven organophosphates in kilograms (kg) | 235,141 | 3721 | 1.6 |
| AGG_HI_LB | The sum of the EPest-high estimates for all seven organophosphates in pounds (lbs) | 518,396 | 8204 | 1.5 |
| HI_LOG_RAT | The log of the concentration of organophosphate use across- harvested cropland for EPest-high estimates, in pounds per square mile | 1455.00 | 2.30 | 0.16 |
| HILBSQMIHC | The concentration of organophosphate unused cross-harvested cropland, for EPest-high estimates, in pounds per square mile | 12,100 | 200.00 | 1.7 |

Source: U.S. Environmental Protection Agency 2020.

2.2.3. Gridded Population Data

Population data were obtained from the 2017 NASA's SEDAC (Socioeconomic Data and Applications Center) project at Columbia University database, which aggregates census block data and residential/household information into latitude-longitude grids at a 30 arc-second resolution [41]. SEDAC data was selected to avoid irregular unit shapes of census units and to achieve high-resolution exposure information that also could be up-scaled to census units [29]. SEDAC enables this study to work outside of census blocks and buffer population over land cover data. The data frame projection was set to the total population using GCS_WGS_1984 and da-tum D_WGS_84. The base map frame was re-projected to NAD83 State Plane West to represent the county's spatial reference more accurately (https://www.earthdata.nasa.gov/eosdis/daacs/sedac, accessed on 21 April 2023).

**Table 3.** Organophosphate Pesticide Usage in Macon County, 2017.

| Pesticide Name | Usage kg/ha |
|---|---|
| Acephate | 24.70 |
| Chlorpyrifos | 50.60 |
| Dicrotophos | 94.30 |
| Phorate | 410.50 |
| Phosmet | 170.30 |
| Terbufos | 150.00 |
| Tribufos | 2843.30 |
| Total | 3743.70 |

Source: U.S. Environmental Protection Agency 2020.

### 2.2.4. Meteorological Data

The office of the NOAA Assistant Administrator for Weather Services (National Weather Service (NWS) Director) database served as the primary source of weather data for the current study. The NWS collects hourly data on surface weather, such as air temperature, wind direction and speed, precipitation, pressure, and humidity. The meteorological data for Macon County covered the entire study period, and all the climatic factors required have been obtained from https://www.weather.gov/wrh/Climate?wfo=bmx, accessed on 3 May 2023.

### 2.3. The Methodology

This study provides an integrated approach of remote sensing, GIS technique, and Ag-DRIFT model that produce thematic information of residents' exposure to organophosphate pesticide drift out from intended targets over Macon County. It integrates the Cropland data set, LULC, available pesticide usage, and population grid data to model population exposure to organophosphate pesticides on both County and field levels. This study estimates the potential exposure to organophosphate pesticides based on Wan's approach [29]. This approach has significant and promising potential for determining exposure to various pesticide components in health and environmental research, especially for the areas that have no pesticide data. The methodology for the current study was modeled from various studies: [27–31]. The methodology has assumed that all croplands in Macon County are using pesticides and that all public and private farms adhere to restrictions concerning the standard quantities of pesticides used. Furthermore, pesticides will drift off-target evenly, taking the primary drivers (wind, followed by humidity and nozzle technology) into account.

Pesticide weight and specific drift potential will be conducted based on the Cropland Data Layer (CDL) dataset multiplied by the amount of pesticide use per area. Then, the MLC's method will be performed to classify the LULC classes. Followed by training data and pixel spectral signatures will be executed to represent more than 55% of Macon County to estimate the accurate pesticide drift. According to Wan, 2015, in the current study, agriculture pesticide drifts exposure linked to the Macon population grid level is calculated using a buffer zone-based exposure drift model. Since the organophosphate pesticide usage data was collected at the county level in Macon, the data was first decomposed onto land use pixels using land use and crop data. A buffer-based exposure model is then used to calculate pesticide exposure at the population grid level.

$$P_{ci} = \frac{C_i W_i}{\sum C_k W_k} Pt \tag{1}$$

where $C_i$ represents the number of pixels for the $i$th crop, $W_i$ represents the average pesticide usage for the $i$th crop, $Pt$ represents the total amount of pesticide used for the entire county,

and $P_{ci}$ represents the total pesticide usage for the crop (*i*) within the county. To convert the population raster grid to points and to calculate the population volume and density, we created, for each cell of the input raster dataset, a point positioned at the centers of cells that they represent. Using raster to point, centroids were created on each cell, creating a point at the cell center with the grid code that equates to the total population value of the cell.

As many GIS-based exposure drift studies rely on pesticides spreading within a particular buffer zone to measure exposure, the accuracy of the buffer zone becomes vital. Establishing a specific buffer zone is complex due to various considerations such as climate conditions, chemical compositions of pesticides, and application techniques [31]. We converted a raster dataset obtained from NASA's SEDAC database to point to features. For each cell of the input raster dataset, centroids were created on each cell, creating a point at the cell center with the grid code that equates to the total population value of the cell.

A buffer zone around each population grid will be established to include all pixels that are identified as belonging to any of the two crops and the total amount of the estimated usage of organophosphate pesticides. For that population grid, the pesticide exposure will be represented by this total amount, indicating pesticide-weighted exposure [29]. To ensure that the middle of the population grid could extend to 750 m, the radius of the buffer zone is set to 1000 m (750 × 1000)/4 given that the average length is approximately 1000 m (shown in Figure 4). In addition, Ward et al. [25] suggested 750 m as an effective range of pesticide drift, and this distance will be used to create the buffer zone. Therefore, a 437.5-m will be created off the pivotal point of each cell. The geodesic method will be utilized, which will maintain an oval shape and be converted into arc seconds. Hence, all the buffer zones with a centroid cell within Macon County will be selected. Zonal statistics were run using the 437.5-m circle to calculate the total crop pixels for each circle.

$$Bz = Pd + Pg/4 \tag{2}$$

where *Bz* is the buffer zone, *Pd* is the mean pesticides drift, and *Pg* is the population grid. It is noteworthy that this equation agrees with [25–31,42–44] studies. A proximity measure, for example, determines that an individual or area unit is exposed to agricultural pesticide drift if its location is within a certain distance of agricultural land; an acreage measure assumes that exposure level is proportional to the total acreage of agricultural lands surrounding the residential location. From this centroid, a 437.5-m buffer was created off the pivotal point of each cell. Utilizing the geodesic method, which maintained an oval shape and was converted into arc seconds. All the buffer zones with a centroid located within Macon County were selected. Zonal statistics were run using the 437.5-m circle to calculate the total crop pixels for each circle, as shown in Figure 4. The population-weighted is designed to 'up-scale' exposure data to area levels is calculated as:

$$Exp = \frac{\sum_j Pop_j Exp_j}{\sum_j Pop_j} \tag{3}$$

where $Pop_j$ and $Exp_j$ are the population size and pesticide-weighted exposure index for the *j*th population grid inside the area unit, respectively. The above formula shows that the area level exposure is estimated as the average exposure of the county population.

The approach of this study follows two levels of estimation, the first is the estimation of potential drift at the county level, and the second level is the estimation of potential drift at the field level. Since the pesticide usage data was at the county level, this data will be decomposed to land use pixels using the land use land cover, cropland, and population gridded data. The potential pesticide exposure drift at the population grid level is calculated using a buffer-based exposure model. On the other side, since the Cropland Data Layer (CDL) dataset provided accurate crop area (ha), and because the rate of pesticide use in Alabama is 2.24 kg/ha [32,33], it was possible to use these data as inputs to the AgDRIFT model to estimate the potential drift of pesticide from their intended targets at the field level as well.

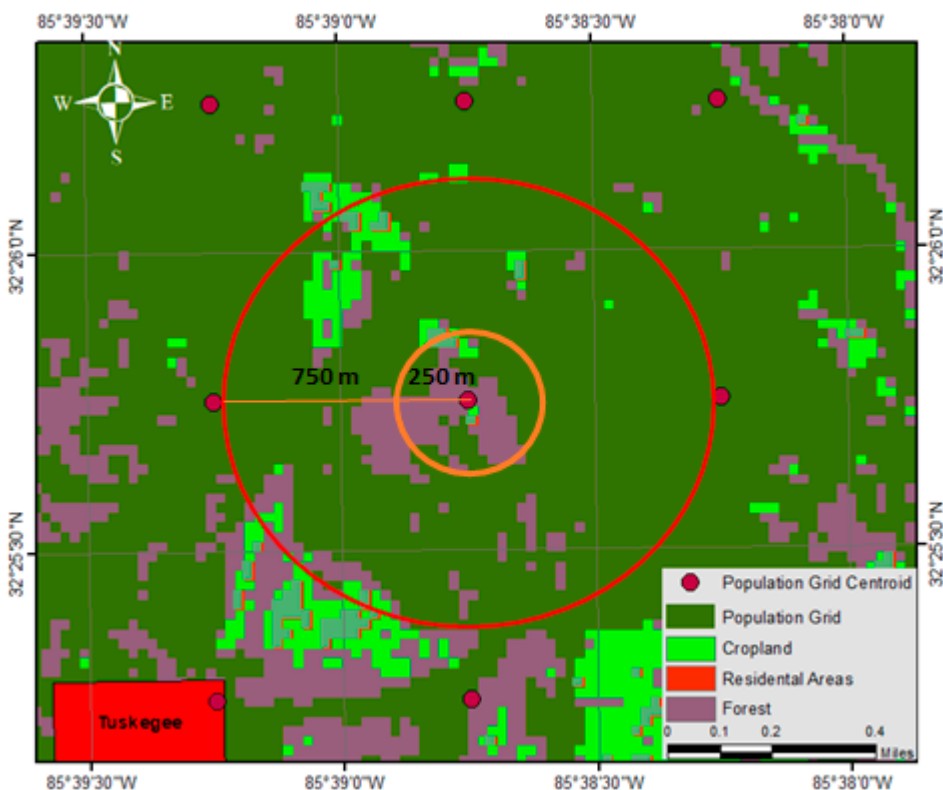

**Figure 4.** Buffer-based pesticide exposure modeling.

### 2.4. Assumptions and Limitations of the Study

The methodology of the current study does not take into consideration the primary variables in drifting pesticide transfer, such as the topography. The wind is the major driver of drift movement, followed by nozzle type, spray pressure, and relative humidity are the key variables. Without taking into consideration any basic factors, this study assumes that pesticides will drift off-target commensurate. The pesticide weight and specific drift potential for the compound were treated as equivalent. Since Alabama does not collect pesticide usage data, the study assumed that all crops in Macon County are treated equally with organophosphate pesticides. For the simulations, this study assumed that the wind is always blowing toward the off-target receiving sensitive resources, it's about 4.47 m/s (10 mph), and the temperature and relative humidity are 30 °C (86 °F) and 50%, respectively.

The main objective of our research is to examine the effectiveness of utilizing remote sensing and GIS techniques to estimate exposure to organophosphate pesticide drift over Macon, Alabama.

### 3. Results and Discussion

Despite the study's limitations, the study's assumptions were mostly reasonable and appropriate for achieving spatial accuracy at the county and field levels. Due to a lack of accurate pesticide data in Macon County, our study developed a method that maximizes the benefits of GIS applications and remote sensing tools in extracting land use land cover data, population distribution data, and pesticide use data at the county and field level to estimate agricultural pesticide exposure drift.

### 3.1. Potential Pesticide Drift Estimation on the County Level

As the total of the 55% stratified sample of spectral signatures used to create training data or regions of interest (ROIs) in 2017, 2.932 km² total were employed. This spectral signature was subsequently utilized to classify 4.390 km² of main LULC classes in Macon County in 2017, with a significant Kappa accuracy assessment of 0.8778. Compared to

squares, which were typically 0.40 km² (mean 0.20), the median size of a Landsat segment was 0.025 km² (mean 0.02). The spatial accuracy was performed on images classified from the classification range, which was derived from the range of Landsat-8 spectral signatures compared with topographic mapping data and the field study for the Province of Macon County.

This study defined the main classes of LULC and calculated the MLC using a data reduction technique enhanced by resizing the spatial resolution of Landsat (30 m) with the Sentinel-2 satellite image (10 m). Our study maximized the benefits of the MLC method by obtaining accurate data from 30 m of the main LULC units for Macon County, as shown in Figure 5, and in a 10 m spatial resolution to distinguish between different classifications such as residential areas, forests, water bodies, and agricultural areas as shown in Figures 6 and 7.

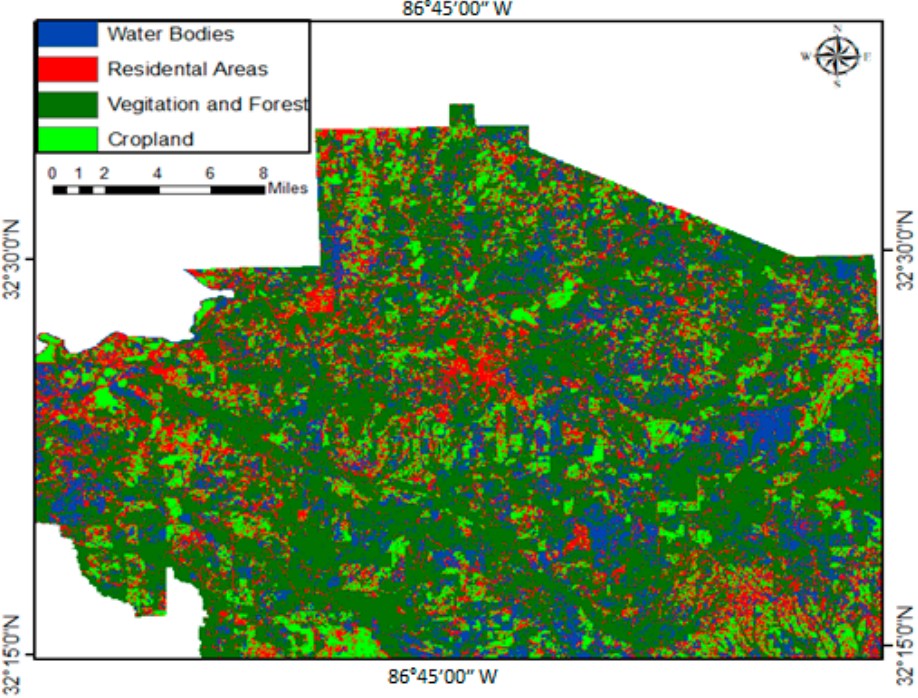

**Figure 5.** Supervised classification of Macon County 30 m.

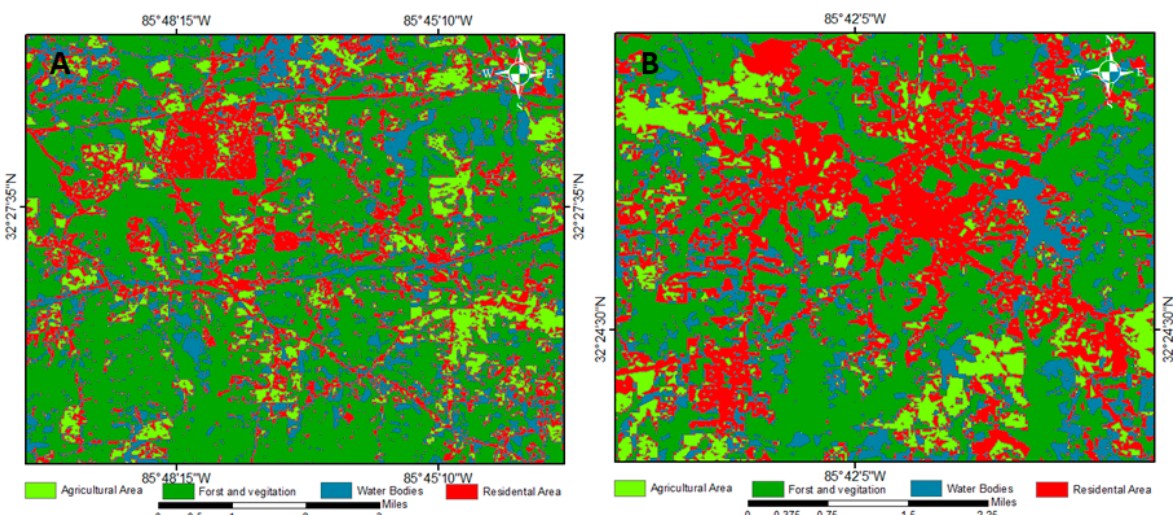

**Figure 6.** (**A**,**B**) Maximum Likelihood Classification of Tuskegee on the right and Franklin in the lift 10 m.

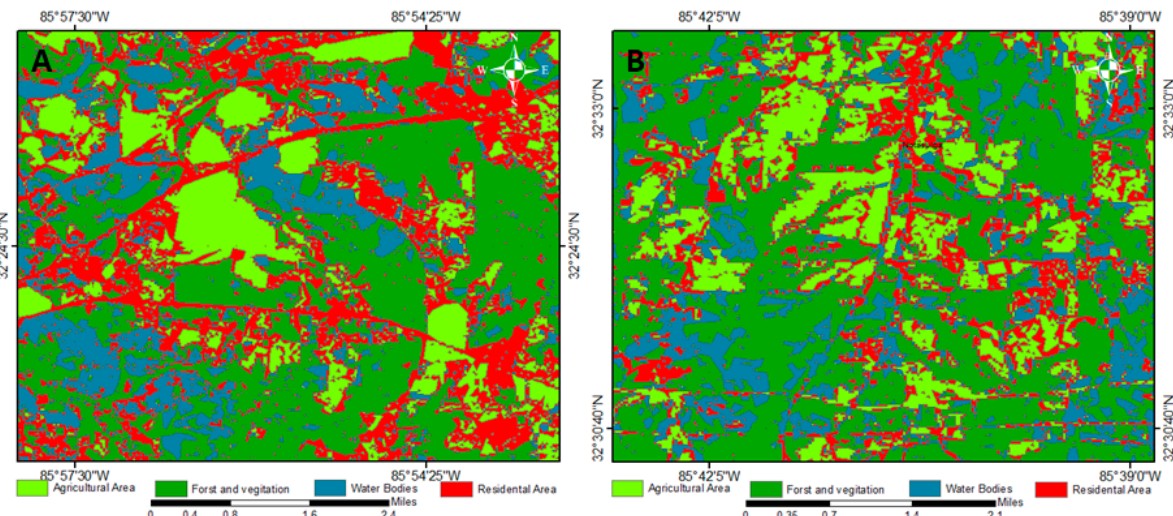

**Figure 7.** (**A**,**B**) Maximum Likelihood Classification of Notasulga on the right and Shorter on the left 10 m.

Figures 6 and 7 depict the classes of land cover categories for the main cities in Macon County with 10 m spatial resolution. Notably, all categories exhibited a great match with ground reality after applying the accuracy assessment, with only a few exceptions for some grid points in the southern margins of Macon County. It is found that agriculture constitutes the predominant type of land use. Due to its flat topography and rich soil, Macon County is perfect for mono-cropping, which is the method used for growing cotton and corn. The water bodies contributed 2.5% of the total Macon County area, and the residential area explains 6.6% of the maximum likelihood classification results. Where the percentage of croplands was about 15.5%, and finally, the forests and vegetation areas represented the majority of Macon County, where they represented about 76%. This percentage can be expected, given the agricultural nature of Macon County. In particular, they often receive larger amounts of pesticides than other urban counties, and accordingly, they are more sensitive to even small amounts of seasonal pesticide applications. We relied on the Cropland Data Layer (CDL) formal dataset shown in Figure 3 to estimate the quantities of pesticide usage in kilogram/ha.

Figure 8 shows the population-gridded data that has been derived and employed to fill the gap in obtaining accurate data, particularly regarding the volume and distribution of the population in Macon County. The spatial pattern of population density is mostly homogeneous to the pesticide usage display, with the population in the central and western regions and Tuskegee experiencing no exposure.

It was found that using the full grid rather than just the circle inside the grid would be a more accurate way to measure exposure after looking at the 437.5-m buffers within each population grid. Much of the agricultural pixels were located just outside the 437.5-m buffer zone in many of the grids. Buffering from the centroid operates under the assumption that the population is centered in the center of the grid, which is problematic given that population might be distributed over a grid. This study found that 62% of the population of rural areas in the eastern and southern parts are exposed to the drift of pesticides within buffer zone 437.5 m, and this percentage increases and reaches 74% when applying the 380-m buffer zone range, as shown in Figure 9.

In detail, the results further indicate that 6.6% of about 1000 residents of Macon County's total population are considered potentially severely exposed, and the affected population resides primarily in rural areas. The results indicate that 23% of about 3340 residents of Macon County's total rural population are considered potentially medium to high exposure. At the same time, 38% of about 5500 residents in a suburban area are considered to have potentially low to medium exposure.

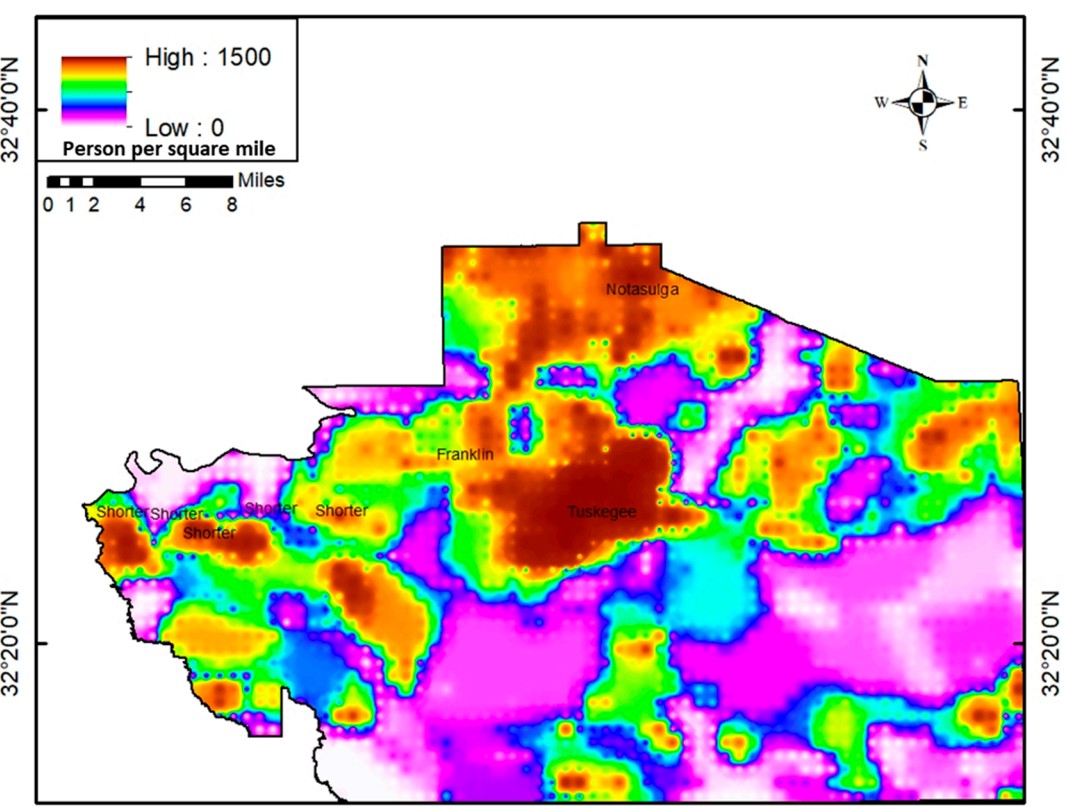

**Figure 8.** Population Gridded data of Macon County(Derived from NASA's SEDAC Map Database).

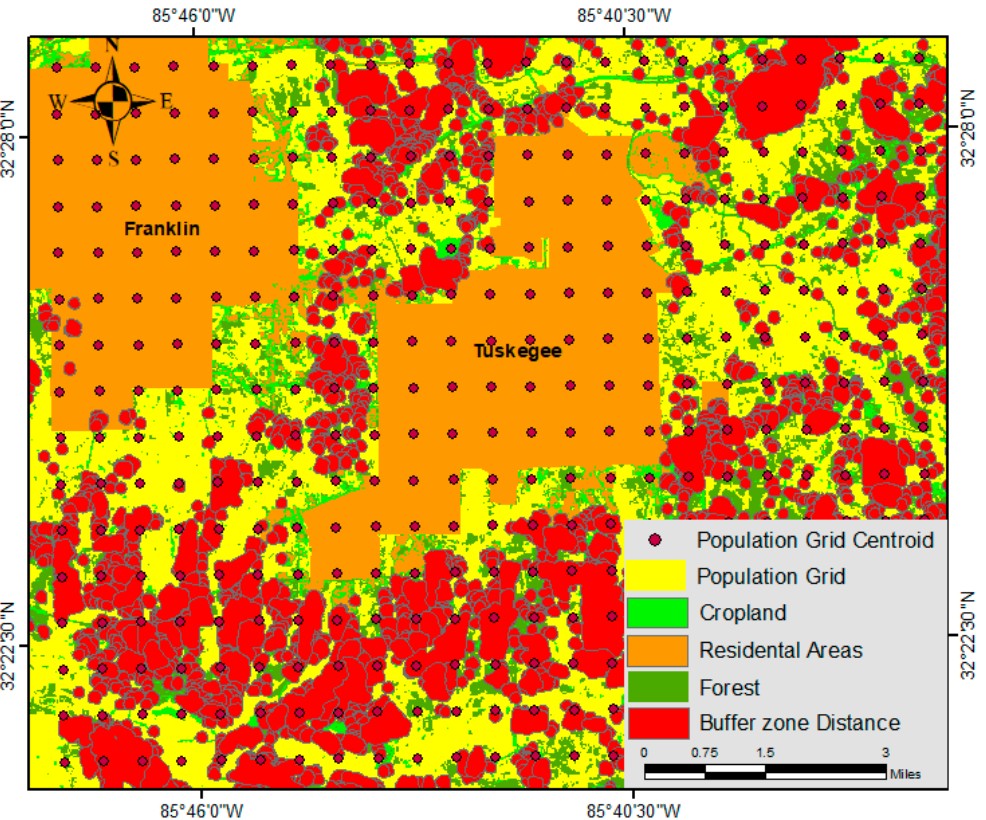

**Figure 9.** Buffer zone-based pesticide exposure and population grid modeling.

Pesticide exposure is almost non-existent in the county's south and east-central regions (which are primarily covered by forest). It can be observed from Figures 10 and 11 that the pesticides used are in close accordance with crop distributions and local geographic characteristics. Macon County is the least populated in the eastern part of Alabama, where more croplands and forests and fewer residential areas occur, with approximately 19,532 persons in 2020. Higher pesticide exposure occurs in the southern and eastern parts of Macon County, whereas the northern part of the county is located close to a main urban area and has considerably less exposure. Residents in Tuskegee City, the county seat and first most populated area, showed very weak exposure to pesticides, as shown in Figure 11.

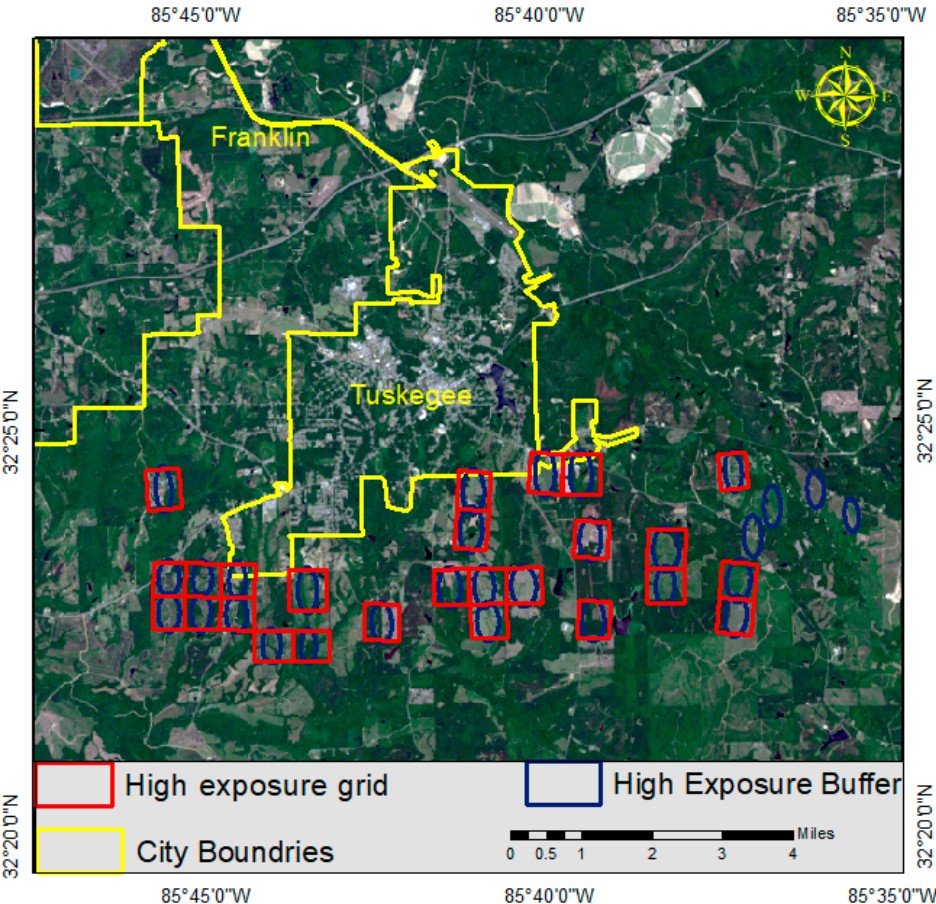

**Figure 10.** The Potential high and low pesticide exposure drifts over Tuskegee.

Figure 11 illustrates the average of pesticide exposure categories over Macon County based on Wan's proposed exposure model, which has been upscaled from population grid exposure data. The population in the middle and northwestern provinces, as well as those in the capital Tuskegee, Notasulga, Shorter, and Franklin, were exposed to moderate-to-weak drift. Accordingly, the current study could estimate the total residential exposure to drift and identify areas of high exposure. Figure 11 Also explains that the southern and eastern boundaries of Tuskegee town have had agricultural areas and thus high exposure to pesticides, but at the same time, have exceeded the maximum limit of the buffer zone. Thus, the risk of exposure to pesticide drift for Tuskegee residents is very weak. Overall, residents nearby the rural-urban agricultural interface have been demonstrated to have a high amount of pesticide drift.

Figure 12 depicts the pattern of Organophosphate Pesticide exposure drift in Macon County, which was derived from exposure information from population grids and based on Table 2 data. It can be concluded that the area under the administration of Macon County reached 38,963 acres, which represents 1.20% of Alabama. It is also clear that acres of

harvested cropland reached 26,305 acres, which represents 1.19% of Alabama. The square miles of harvested cropland in Macon County in 2021 reached 41 SQ_MI_HC.

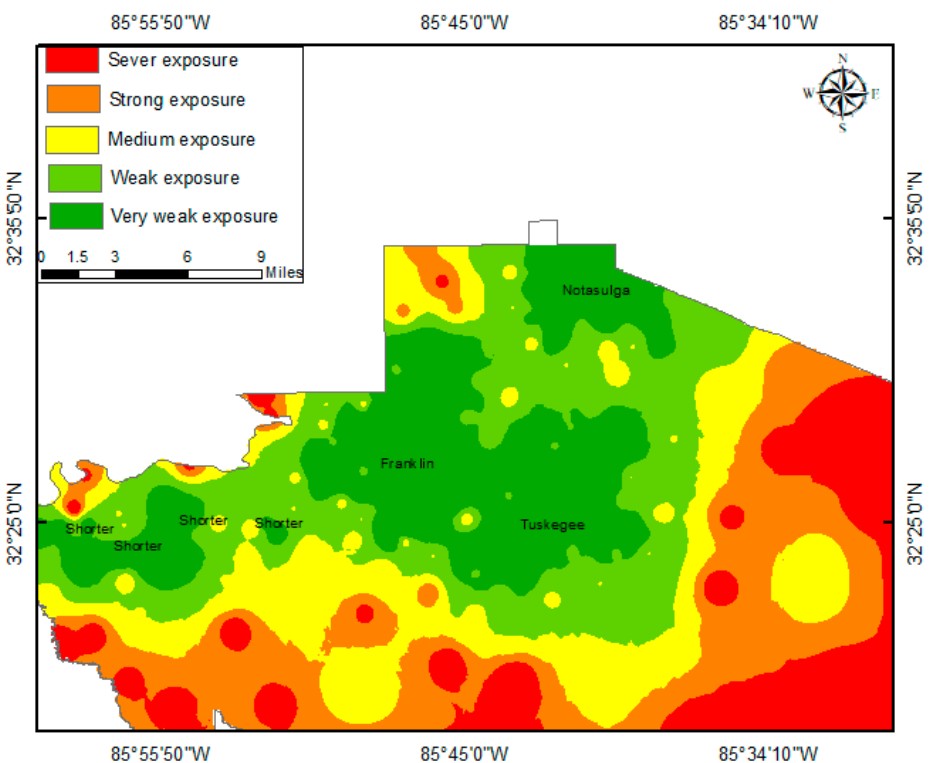

**Figure 11.** The average of pesticide exposure categories over Macon County.

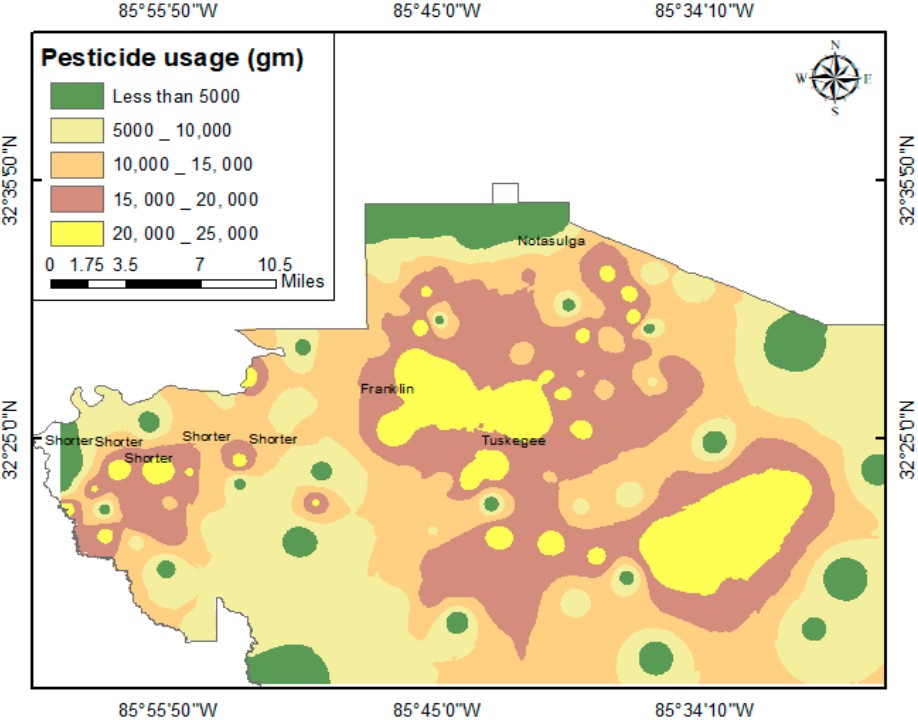

**Figure 12.** The Pesticides Usages (gm) over Macon County in 2017.

It can be observed that the log of the concentration of pesticide uses across harvested cropland for EPest-high estimates for Macon County is 2.3 kg for EPest-high estimates, and

the pesticide concentration across harvested cropland was 199.61 pounds per square mile. The sum of the EPest-high estimates for pesticides is 3721.66 kg. The sum of the EPest-high estimates for pesticides in pounds (lbs) reached 8204.72. In the urban agricultural interface of the county with high levels of exposure drift, those with average exposure greater than 25 kg/ha were concentrated. Urban areas had substantially lower exposure risk levels lower than 10 kg/ha.

### 3.2. Potential Pesticide Drift Estimation on Field Level

The regulatory Authorities in North America, Europe, and other regions rely on spray drift models to estimate the amount of pesticide exposure drift from aerial or ground applications [21]. The AgDRIFT model was used to generate deposition curves for ground and aerial spray on the field level, as shown in Figure 13. AgDRIFT provides estimates of pesticide spray drift based solely on distance downwind as well as the effectiveness of the spray equipment. It was created to evaluate different spray drift conditions. To quantify the potential exposure, the AgDRIFT model (developed by the Spray Drift Task Force) was used. The use of a flexible drift model and accurate land cover information allows the estimation of potential drift for individual ponds spatially distributed over a large area.

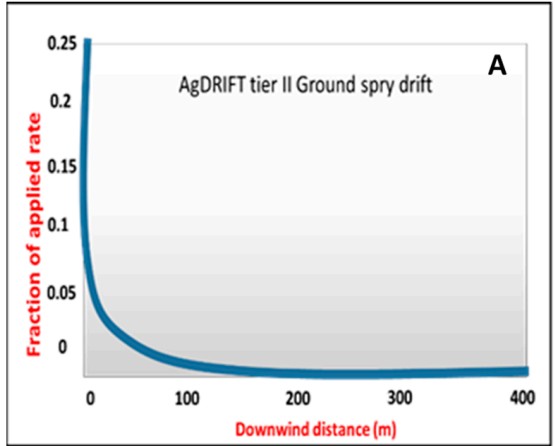 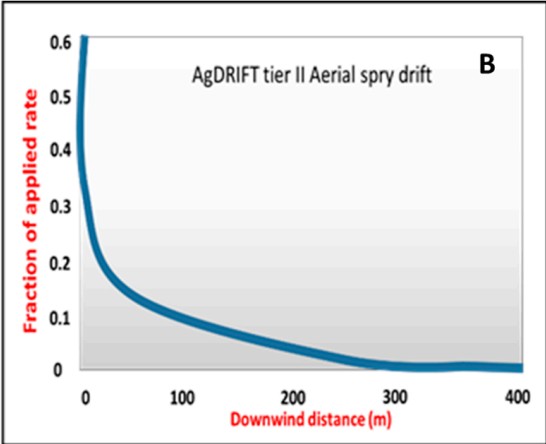

**Figure 13.** Estimation of spray drift deposition of agriculture pesticide; (**A**) Ground-boom, and (**B**) Aerial application generated by the AgDRIFT model; the Y axis is the fraction of applied rate per unit area of deposition.

Five simulations were conducted in the largest fields and farms in Macon County from 1 August 1 to 29 August 2017, using a total of 130 kg of organophosphate pesticides, mostly on cotton and corn fields. Seven organophosphate pesticides were identified as having potential for spray drift, considering the proximity of the applied field to the sensitive resources as well as the wind directions on the application day. A total of five pesticide applications simulations were made applied within the 437.5-m buffer area Table 4. Naturally, five pollution roses were applied to illustrate the frequency distribution of wind direction temporally related to the number of pesticides applied in the five fields.

The simulations were distributed geographically within Macon County, the fields were selected in the North, South, East, and West. The number of pesticides drifted on the sensitive resources ranged from 2.99 to 6.02% of the total amount that was applied on the five fields (or 129.65 kg of pesticides) for the ground and aerial spray application, as shown in Table 4 and Figure 14. The simulation of the pesticide drift in Macon County remains a challenge due to a lack of data, application timing, and pesticides amount by various crops. However, the approach of the current study enabled it to obtain the number of pesticides used based on the accurate area of crops multiplied by the average requirement per hectare. As for the timing of application, the five simulations were distributed overtime

during the entire month of August, which is the climatically appropriate month for the application of pesticides on summer crops, especially in Alabama.

**Table 4.** Drift associated with the application of Organophosphate pesticides to crops in five selected fields in Macon County.

| Field Name | Spray Date | Location Lon/Lat | Pixel Count | Area (ha) | Applied Rate (kg/ha) | Wind Direction | Method | Drift Mass (kg) | Drift/ Total Applied (%) |
|---|---|---|---|---|---|---|---|---|---|
| Big Creek | 1 August 2017 | −85°45′00″ 32°42′06″ | 41,656 | 92,643 | 2.24 | SSW | Aerial | 1.20 | 6.02 |
| Dicks Creek | 7 August 2017 | −85°59′40″ 32°24′50″ | 60,255 | 134,004 | 2.24 | NNW | Ground | 0.90 | 2.99 |
| Persimmon Creek | 15 August 2017 | −85°54′53″ 32°32′42″ | 41,745 | 92,838 | 2.24 | SSE | Aerial | 1.20 | 5.75 |
| Long Branch | 22 August 2017 | −85°48′52″ 32°24′48″ | 41,540 | 92,381 | 2.24 | SE | Ground | 0.90 | 4.35 |
| Shorter | 29 August 2017 | −85°33′00″ 32°39′09″ | 75,075 | 166,963 | 2.24 | WNW | Aerial | 1.20 | 3.21 |

The rates of applied pesticides were about 21 kg/ha for the fields of Big Creek, Long Branch, and Persimmon Creek. While the rates were about 30 and 37 kg/ha for the fields of Dicks Creek and Shorter, respectively. The pollution (pesticide) rose diagrams in Figure 14 were constructed as follows. The wind speed, frequency, and direction were analyzed for application days of all five fields for a total of 5 application days. The pesticide roses were investigated mainly for cotton and corn, with simulations carried out using both aerial and ground applications.

This simulation was carried out for the worst-case scenario. The pesticide rose was divided into eight quadrants: N, NE, E, ES, S, SW, W, and NW which equals 0, 45, 90, 135, 180, 225, 270, and 315 degrees respectively. The potential drift deposition values were obtained after running the AgDRIFT model. This analysis was restricted to the AgDRIFT input parameters used by the United States Environmental Protection Agency (USEPA) in conventional regulatory drift estimation. As for estimating the amount of pesticide drifting from their intended targets, It was calculated based on the prevailing wind direction and its frequency relative to the pesticide total applied, which was calculated based on the multiplication of the applied rate (kg/ha) and actual cropland area (ha).

The findings indicate that winds with speeds >4.47 m/s (10 mph), or those more likely to result in increased off-target drift, could be significantly associated with just one or a few directions than slower-speed winds for a given site. Due to the larger drift load events, it would be assumed that wind direction would become an even more relevant variable.

Cropland Data Layer (CDL) dataset was used to analyze the spatial relationship between crop areas (cotton and corn) and the sensitive resources. The remote sensing techniques provided accurate data input for the AgDRIFT model, allowing for the quantification of potential drift. The estimation of potential drift for all fields that were spatially dispersed over a vast area was accomplished using the AgDRIFT model and reliable LULC data. There were 84 static water bodies spread across the study area's 50,000 acres, of which 45 were within the spray drift range of cotton and corn. Overall, the five fields had roughly 311 houses covering a land area of 5800 km$^2$ under potential high pesticide exposure drift. Approximately 175 acres of sensitive crops are subject to high potential exposure drift from pesticides.

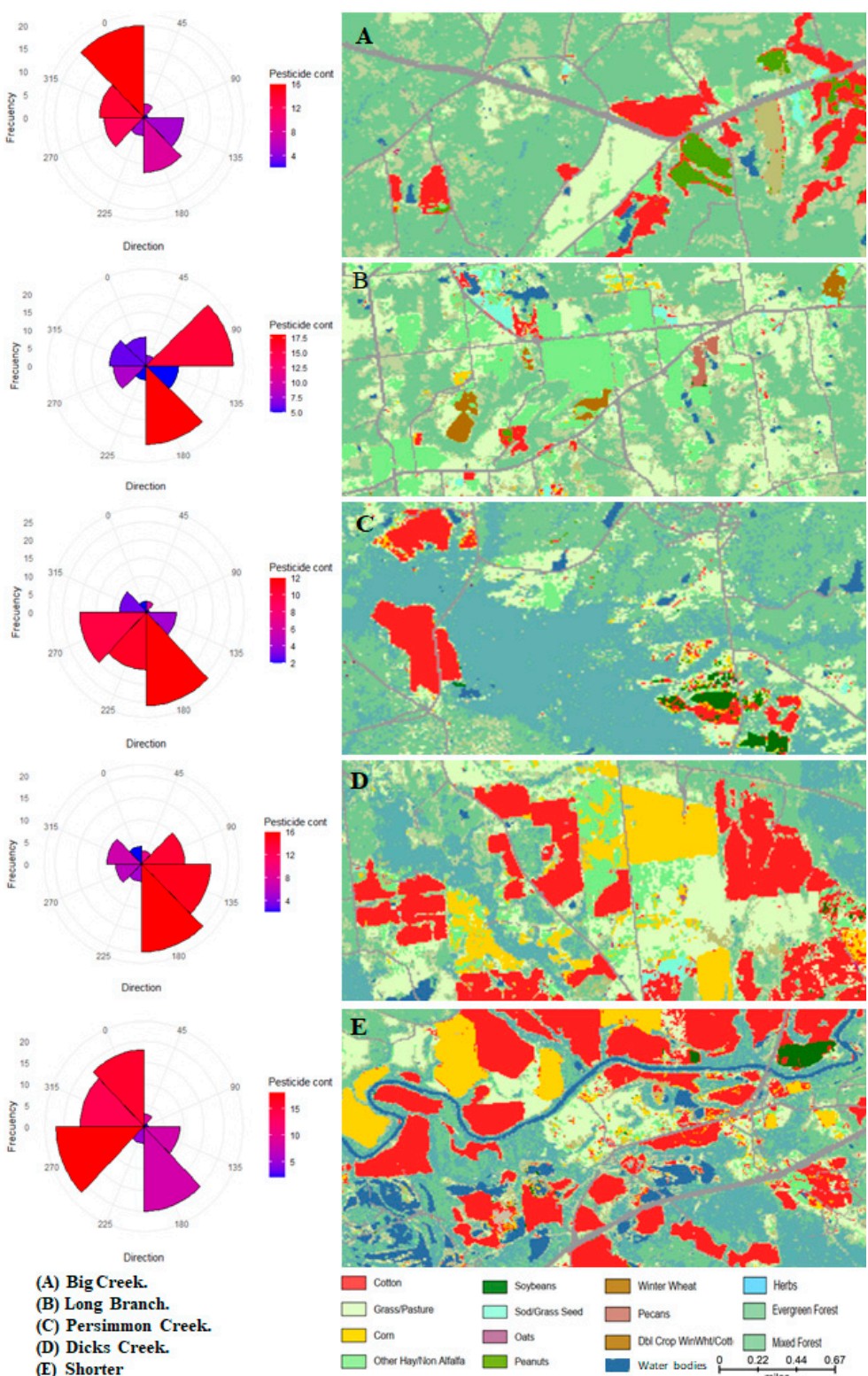

**Figure 14.** Drift simulation associated with the application of Organophosphate pesticides to crops in five selected fields in Macon County; (**A**) Big Creek; (**B**) Long Branch; (**C**) Persimmon Creek; (**D**) Dicks Creek and (**E**) Shorter Town.

Identifying population exposure drift to agricultural pesticides is also critical for determining the adverse health consequences of exposure, especially for the people living in urban agricultural interface communities. Recent healthy studies have demonstrated that

agricultural workers, applicators, and rural residents had double the amount of pesticide residue in their residences [25,26,29]. The danger of both acute and long-term exposure to pesticide drift makes it crucial to address this issue [30]. The use of organophosphate pesticides by agricultural workers in Macon County, along with a lack of pesticide data use, makes it challenging not only to estimate pesticide drift to residential areas but also to determine the adverse health impacts of this exposure. To address this challenge, this study utilized an approach that employs GIS, land use, and land cover data derived from remote sensing, population gridded data, and county-level pesticide usage data to estimate population exposure to organophosphate pesticides. The current study is considered one of the first in Alabama that assesses high-resolution patterns of pesticide exposure drift at the state and field levels.

This study has shown an effective procedure for Utilizing Remote Sensing and GIS techniques to estimate the exposure to organophosphates and pesticide drift based on Wan's approach. This approach allowed for a study of Macon County by assuming that a population would potentially be exposed to high spray drift of pesticides if a threshold of nearby cropland was fulfilled. This approach is necessary nowadays to estimate potential spray drift from pesticide uses with appropriate spatial resolution in states that do not have a database of pesticide uses. The Cropland Data Layer (CDL) dataset allowed us to determine exposure at high spatial resolution by defining the crop area with high accuracy, it was thus possible to calculate the amount of applied pesticides. The use of SEDAC population-gridded data allowed us to examine exposure drift in a high spatial resolution that is higher than that of census units.

The current study illustrated that the potential exposure to pesticide drift was concentrated in rural areas with low population densities in Macon County. The potential exposure to pesticide drift is further reduced away from rural areas, especially near major cities in the center and west. Wind speed, nozzle type, spray pressure, and relative humidity are the key variables that affect spray drift. However, variables such as wind and application methods may increase the possibility of pesticides drifting from their intended target. Moreover, from a climatic perspective, it is necessary to stress that pesticide usage data availability is not the unique driver that controls the spray drift out of the intended fields, with implications for assessing the environmental and health of pesticide impacts over Macon County. The pesticide spray drift was heavily influenced by wind patterns; thus, buffer zones should be modified to account for any additional possible spread. The majority of pesticide labels instruct applicators not to spray when wind speeds are greater than 10 mph or when areas of vulnerability are downwind. However, the wind direction and meteorological conditions, which are important factors in any study on the drift, nevertheless were not considered in either Ward's [26] or Wan's [29] studies.

This study demonstrated that Macon County is a mix of rural and urban residents, making it well suited to determine how many of the county's residents are at high exposure and to examine the spatial pattern of potential population grid exposure to pesticide drift. The analyzes indicated that the pesticide usage in Macon County showed high-usage areas were mostly concentrated in the county's southern and central regions, which corresponded to the main cropland concentration areas. Albeit with their coverage of more than 25.000 kg of agricultural pesticide usage, the characterization of Tuskegee, Notasulga, Franklin, Woodland, and Shorter in the high-usage pesticide category is not well understood from a practical perspective. This is mainly due to the uneven geographical distribution of pesticide information over time and space, particularly at more residential sites, and also attributed to high population density and the long distance of cropland away from those cities.

Previous studies estimating pesticide exposure drift observed consistent results on the appropriateness of Wan's method for estimating exposure to pesticide drift for states that do not have a database of pesticide uses. It is worth noting that the entire approach is easily applicable to exposure spray drift estimation at the County level. The results of the current study agreed with the results of previous studies that relied on the Wan

approach. [29] Concluded that the total area of cropland had a greater impact on pesticide drift exposure than proximity studies, and they concluded that a population surrounded by 68.7 percent of cropland would have significant exposure. In another study, [31] it was found that areas of higher people density have much less potential drift exposure than rural ones do. It was found that crops were observed within a 402-m buffer zone in 88 percent of the population buffers. When highly populated and metropolitan regions were omitted, it became apparent that 13.57% of the population in Madison County, Illinois, satisfied the threshold for high exposure, as opposed to the 3.3% who were judged to be highly exposed.

The current study accords with [25,27,29] studies that rural areas are at a higher pesticide exposure drift, indicating the necessity for a more accurate examination of rural areas. Future research should focus on isolating rural areas and estimating pesticide exposure drift based on the actual number of rural populations. Estimating the exposure drift at the county and state levels is unreliable when non-rural areas are considered, and it reduces the potential impact of exposure outcomes. The lack of a suitable model for estimating pesticide drift in Alabama motivated the work team to develop previous research attempts and a model that is compatible with the residential environment and weather conditions of Macon County, and previous research backs it up.

The simulation of the five experiments carried out at the field level confirmed the outputs based on the [29] approach, which was carried out at the level of Macon County. Our study provided an estimate of agricultural pesticide drift on both County and field levels, which can be used to identify probable locations of high exposure. Highlighting high-risk geographic locations can help identify possible areas of concern. This conclusion is consistent with [25–31] results; it could be the first study of the Southeastern Counties that can highlight potential geographic areas of high exposure drift. This study also demonstrated the effectiveness of utilizing remote sensing and GIS techniques to estimate agricultural pesticide exposure drift, notably in regions that do not have accurate data on the number of pesticides used. Albeit with the current study limitations, it provided a real and accurate assessment of pesticide exposure drift. This methodology should serve as a framework to inform more detailed studies that combine the trends of drift. According to this study's results, we have demonstrated that the integration between GIS and remote sensing techniques has allowed us to obtain logical potential estimates of pesticide drift at both County and field levels based on Wan's approach, which would enable us to fill the lack of pesticide use data in most US states and develop high spatial resolution of pesticide data affected by minor errors.

## 4. Conclusions

Very few states, except California, have full pesticide use reporting databases that provide sufficient temporal and spatial precision to accurately estimate the impacts of pesticide spray drift on the population. As a result, this study provided an extensive method for estimating high-resolution information on organophosphate pesticide exposure at the county and field levels. Based on the integration of different datasets (i.e., Cropland Data Layer, population gridded data, LULC, and GIS technique) to overcome the lack of pesticide data and improve data coverage and comparability, this approach allowed us to provide an estimate of how much of Macon County is potentially exposed to organophosphate pesticide drift.

The suggested approach provided more comprehensive data regarding pesticide exposure drift than traditional exposure models and indicated significant potential in addressing pesticide-related environmental and health issues. Wan [29] examined pesticide exposure without specific pesticide data using pesticide exposure modeling based on GIS and remote sensing land use data. This approach was validated for estimating population exposure to pesticide drift outside of the intended targets. In this study, remote sensing and GIS tools provided accurate spatial data on potential pesticide exposure drift. It can be concluded that the overlay of the Cropland Data Layer (CDL) dataset, SEDAC population gridded data, and pesticide usage data with LULC was remarkably useful for identifying

residential areas in Macon County with potentially high risks of exposure to pesticide drift. Despite its limitations, this study provided an inclusive estimation of exposure drift at the county and field levels. This estimation can help public health professionals, and regulators protect human health and enforce mitigating measures.

The authors speculate that the results obtained based on regulatory assumptions are specific to the current study and that the results of estimating insecticide drift will inevitably change due to the impacts of other climatic factors such as (relative humidity, temperature, wind speed, and direction). This study benefited from the findings of previous studies and added that the potential estimate of pesticide drift from its intended targets is closely related to the local prevailing weather conditions at the time of application.

**Author Contributions:** G.E.A., H.I., S.F. and R.A. have made a substantial, direct, and intellectual contribution to the work and approved it for publication. All authors have read and agreed to the published version of the manuscript.

**Funding:** This work is funded by the 1890 Capacity Building Grants Program (CBG) [Grant No. 2020-38821-31084/project accession No. 1021820] from the USDA National Institute of Food and Agriculture.

**Data Availability Statement:** The datasets generated during and/or analyzed during this study are available from the corresponding author upon reasonable request.

**Acknowledgments:** The authors would like to thank the USDA National Institute of Food and Agriculture for Support and financing.

**Conflicts of Interest:** The authors declare no conflict of interest. The funders had no role in the design of the study; in the collection, analyses, or interpretation of data; in the writing of the manuscript; or in the decision to publish the results.

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
