# Peer review of "Effectiveness of Utilizing Remote Sensing and GIS Techniques to Estimate the Exposure to Organophosphate Pesticides Drift over Macon, Alabama"

_agronomy, doi:10.3390/agronomy13071759_

Round 1

Reviewer 1 Report (New Reviewer)

Line

Comments 

305

Duplicate information in line 305 & 390

155

It is best to define the spatial boundaries of the study area from longitude to longitude And from latitude to latitude

229

The researchers did not explain that the coordinate system for agricultural data is Albers_Conic_Equal_Area

From link: https://nassgeodata.gmu.edu/CropScape/

Additionally, the spatial reference was: D_North_American_1983

And in his article they used D_WGS_84

The researchers did not report what are the transfer parameters

229

The reference should be more specific

(www.earthdata.nasa.gov).

For example:

https://www.earthdata.nasa.gov/learn/pathfinders/gis-pathfinder/geospatial-services

303

The figure dimensions are not good, and there is no coordinate grid

377

In Figure 6, the coordinate grid is not clear

380

In Figure 7, the coordinate grid is not clear

380

Figures 5 and 6 show a legend of different colors and names

389

Figure 8 did not show the area unit for population density

399

The researchers did not explain how to obtain the numbers through tables or figures

406

Figure 6 is not clear, Please number the figures as (a) (b) (c)

Then explain each part what it

418

In Figure 11 please make the scale interval 1 mile instead of 0.8 mile

468

Figure 13, Please number the figures as (a) (b) Then explain each part what it

471

The number of decimals is exaggerated, especially since we are talking about  a 129.657696 kg

479

Clarify the coordinates Long/ lat Is it a decimal degree?

559

The citation is not  similar to the template adopted in the journal

Author Response

Dear Reviewer:

We want to express our sincere thanks for taking the time to review our manuscript. Your efforts and valuable feedback have helped us improve the quality of our work. Based on your recommendations and constructive comments, the authors have made the necessary corrections and modifications to the original manuscript. To make it easier to identify these changes, we have used the "Track Changes" function and highlighted the revised sections in red with strikethrough texts. Please see below for more details.

Sincerely,

The authors

Reviewer 2 Report (New Reviewer)

Dear authors,

After reviewing the article titled “Effectiveness of Utilizing Remote Sensing and GIS Techniques to Estimate the Exposure to Agricultural Pesticides Drift over Macon, Alabama”, proposed for possible publication in the journal Agronomy would like to state that from my perspective, this work presents a topic that is relevant from an environmental point of view. However, I believe that some aspects should be considered and corrected before the work could be considered for publication. I list some suggestions for improvement and details that I observed in it:

The title is not appropriate since it does not reflect the content presented in the manuscript. It should not generalize to pesticides when in fact only the class corresponding to organophosphates is being addressed.

In the introduction, line 38, even though it seems obvious not to use abbreviations without having indicated their meaning at least once (e.g. USA, USGS line 122), apply this observation to the content of the entire document.

Line 99 homogenize US or USA.

The legends inside the figures must use the same font as in the text.

In Table 3, homogenize the names so that they all correspond to active ingredients, change DICROTOP to Dicrotophos and CHLORPYR to Chlorpyrifos.

It is suggested to change figure 3 to individual figures and move them to supplementary material since in the current presentation the information shown is not appreciated and although the quality of the image will improve, according to my perception the problem would continue to be persistent.

Figures 5, 6, 7, 9, and 10 can be enlarged to a larger size since the most significant contribution of the study has to do with the novelty of the application of image processing to estimate exposure to organophosphate pesticides. The images needed to be visually clearer, it is suggested to present these on a sheet each individually and at the highest possible resolution.

Authors should standardize the use of the comma as a mile separator throughout the document.

The results are only described basically without the comparison with similar studies or demonstrating the advantages of this methodological proposal with others already existing and that was mentioned in the introduction.

Authors should standardize the use of the comma as a thousand’s separator throughout the document.

The results are only described basically without the comparison with similar studies or demonstrating the advantages of this methodological proposal with others already existing and that were mentioned in the introduction.

The discussion needs to be improved.

The proposed model should have been statistically validated to give greater certainty and support that it works and meets the stated objective.

It is suggested to homogenize the format of the references in accordance with the guidelines of the journal, in some they place the full name of the journal, and in others abbreviated.

I hope that my comments and suggestions help to improve the work done.

Kind regards

The writing and syntax of the English language must be reviewed since in some parts of the document it is not expressed correctly.

Author Response

Dear Reviewer:

We want to express our sincere thanks for taking the time to review our manuscript. Your efforts and valuable feedback have helped us improve the quality of our work. Based on your recommendations and constructive comments, the authors have made the necessary corrections and modifications to the original manuscript. To make it easier to identify these changes, we have used the "Track Changes" function and highlighted the revised sections in red with strikethrough texts. Please see below for more details.

Sincerely,

The authors

Round 2

Reviewer 1 Report (New Reviewer)

The manuscript has been sufficiently improved to warrant publication in Agronomy

Author Response

Response to Reviewer 1 Comments

1) The manuscript has been sufficiently improved to warrant publication in Agronomy.
Response:
Thank you so much, we would like to express our gratitude and thanks to the reviewer for the positive feedback and helpful comments that supported and enrich these revisions. This result would not have been possible without the insightful notes of the reviewer.

Reviewer 2 Report (New Reviewer)

Dear authors,

After reviewing the article titled “Effectiveness of Utilizing Remote Sensing and GIS Techniques to Estimate the Exposure to Agricultural Pesticides Drift over Macon, Alabama”, proposed for possible publication in the journal Agronomy would like to state that from my perspective, this work presents a topic that is relevant from an environmental point of view. However, I believe that some aspects should be considered and corrected before the work could be considered for publication. I list some suggestions for improvement and details that I observed in it:

Introduction line 41, pesticides refer to insecticides?

The conclusions are not adequate and are interspersed with a bit of discussion that should go in the corresponding section.

I hope that my comments and suggestions help to improve the work done.

Kind regards

The writing and syntax of the English language must be reviewed since in some parts of the document it is not expressed correctly.

Author Response

Response to Reviewer 2 Comments

1) Introduction line 41, pesticides refer to insecticides?
Response:
We are grateful for the valuable note, the term has been corrected to be insectcides in revised manuscript, because the pesticide is the general term for a chemical that kills pests. Pests can be weeds, insects, nuisance rodents, diseases, etc. An insecticide is a type of pesticide. (https://www.llojibwe.org/drm/greenteam/pesticides_Article.pdf).

2) The conclusions are not adequate and are interspersed with a bit of discussion that should go in the corresponding section.
Response:
Thanks for this valuable suggestion, the conclusions section has been rewritten, and a bit of discussion that has moved in the corresponding section.
We greatly appreciate the reviewer's time and efforts, which enable us to improve our work to the highest possible quality We would like to express our respect and appreciate the keen interest of the reviewer in improving the manuscript results and conclusion. This result would not have been possible without the insightful notes of the reviewer.
We would like to take this opportunity to express our gratitude and thanks to the reviewers for the positive feedback and helpful comments that supported these revisions.
Thank you for your consideration of this manuscript.
Sincerely,
Dr. Gamal El Afandi
Tuskegee University, AL, USA

Round 3

Reviewer 2 Report (New Reviewer)

Dear authors,

Thank you for attending to the suggestions and comments for improvement made to the document. I believe that it is now ready to be considered for publication.

Kind regards

The manuscript requires a slight refinement of the language to be ready for publication.

This manuscript is a resubmission of an earlier submission. The following is a list of the peer review reports and author responses from that submission.

Round 1

Reviewer 1 Report

This paper deals with the development of a methodology combining remote sensing and GIS to predict potential exposure of residents of the Macon county, Alabama, USA.

1)     The methodology combining the estimation of land use, pesticide sales and population maps is interesting however the paper does not really demonstrates the novelty of such approach. Resident exposure is a worldwide concern and there are probably equivalent studies that were already published in the US?

 2)     Another weak point was found on the risk model that sems irrespective to teh chemical toxicity or the application technique. The paper does not seem to rely on the risk assessment achieved during the registration process of pesticides by EPA and the resulting recommendations in terms of buffer/safety zones during the application. This evaluation considers spray drift models (and post application volatilization) and related impacts in terms of resident dermal and inhalation exposure. The relationship with the model defined here is not obvious or clear. For example “Pd” in Equation 3 is a distance of 400m that is totally irrespective from the chemical toxicity or the application technique. This appears as a (too) simplistic model. Are there realistic arguments not to consider those models as a basis for this risk assessment ? The way low or high risks for residents are obtained in this paper is not clearly defined.  

3)     It seems there is a confusion between the exposure risk linked to the proximity of cultivated fields and the exposure level at a certain distance (spray drift values at a certain distance)…

4)     Most of the results correspond to qualitative information through maps but global quantitative information is missing. For example, it would have been interesting to get the percentages of the population facing different levels of potential risks.

5)     Isn’t it somehow paradoxical to refine crop surface data at higher resolution (10m – 60 m) and to definition of buffers zones of 400 or 750 m…

6)     This paper mainly considers the proximity of cultivated land beside residential areas as the main factors. Main wind direction is slightly evoked in the paper but does not seem to be part of decision support system.

Detailed comments

Page 3 line 128 – 136 : it is not clear whether EPA data/models (sedimentation spray drift, airborne drift, dermal and inhalation resident exposure,…) are considered in this study.

Page 3 line 138 : the “potential” exposure…  

Page 4 line 167: this study focuses on most used agricultural pesticides… it might be interesting to get this list  as well as a global view of crop type distribution. It is not obvious to understand whether cotton or corn are a majority?

Page 6 Table 2 and Table 3. The case of organophosphate pesticides are chosen is not explicit. Because of their toxicity? Because they represent the main sales? Because they are significantly recovered in water streams? … please explain.

Page 8 line 254 : On which basis, this 400m buffer zone was defined exactly ?

Page 8 line 257 : Is this 750 spray drift distance related to all types of spray application techniques or only to aerial ?  

Page 8 line 266: On which basis this 437.5 m buffer was defined?

Page 11 Line 325-327: The assumption that there a risk of exposure below 400m and no risk above 400m is a bit simplistic. This does not take into account the exposure level at this distance that linked to the product toxicity and the application technique

Author Response

Dear Reviewer:

We would like to express our profound gratitude to you for considering/reviewing the manuscript. We greatly appreciate your time and efforts, which enable us to improve our work to the highest possible quality. Based on your recommendations and constructive comments, the authors have corrected/modified the original manuscript. The revised parts of the manuscript are marked up using the “Track Changes” function”, which highlighted the modified parts in the MS using red-colored and strikethrough texts. Hereunder is the report containing our responses in detail.

Sincerely,

The authors

Reviewer 2 Report

The research design is flawed, the analysis is underdeveloped, and the quality and reliability of the results are poor. 

Multiple sets of land use and land cover data are available worldwide. Most of them are free, high-resolution, and updated often. The US has the NLCD, a good data source for domestic analysis. CDL published by USDA has the crop distribution information. Both NCLD and CLD are high-quality data, so why do you conduct supervised analysis to classify LUCC by yourselves, and what are the accuracies of your work?

Pesticide air-borne drift is highly affected by wind, temperature, and precipitation. Simply assuming an even drift is not acceptable, which is the biggest flaw of your work. 

There are lots of fallacies and mistakes throughout the ms. I am sorry I have to reject your work.  

Acceptable. 

Author Response

(The authors gave the same response as above.)

Round 2

Reviewer 1 Report

The paper was significantly imporved since authors have addressed all comments. 

Reviewer 2 Report

The ms has serious flaws. Although some concerns have been addressed in your edits, more experiments are strongly needed. 

Acceptable.